# Epistasis mediates the evolution of the receptor binding mode in recent human H3N2 hemagglutinin

Ruipeng Lei [1,14], Weiwen Liang [2,14], Wenhao O. Ouyang[1,14], Andrea Hernandez Garcia[1,14], Chika Kikuchi [3,4], Shengyang Wang[3,4], Ryan McBride[3,4], Timothy J. C. Tan [5], Yuanxin Sun[6,7], Chunke Chen[6,7], Claire S. Graham[1], Lucia A. Rodriguez [1], Ivana R. Shen[1], Danbi Choi[1], Roberto Bruzzone[2,8,9], James C. Paulson [3,4], Satish K. Nair [1,5,10], Chris K. P. Mok [6,7,11,12] ✉ & Nicholas C. Wu [1,5,10,13] ✉

The receptor-binding site of influenza A virus hemagglutinin partially overlaps with major antigenic sites and constantly evolves. In this study, we observe that mutations G186D and D190N in the hemagglutinin receptor-binding site have coevolved in two recent human H3N2 clades. X-ray crystallography results show that these mutations coordinately drive the evolution of the hemagglutinin receptor binding mode. Epistasis between G186D and D190N is further demonstrated by glycan binding and thermostability analyses. Immunization and neutralization experiments using mouse and human samples indicate that the evolution of receptor binding mode is accompanied by a change in antigenicity. Besides, combinatorial mutagenesis reveals that G186D and D190N, along with other natural mutations in recent H3N2 strains, alter the compatibility with a common egg-adaptive mutation in seasonal influenza vaccines. Overall, our findings elucidate the role of epistasis in shaping the recent evolution of human H3N2 hemagglutinin and substantiate the high evolvability of its receptor-binding mode.

Since the 1968 "Hong Kong flu" pandemic, H3N2 subtype of influenza A virus has been circulating in human for over half a century. As the major antigen of influenza virus, hemagglutinin (HA) glycoprotein is the fastest evolving protein in human H3N2 virus[1,2]. HA facilitates virus entry by engaging the sialylated glycan receptor and mediating virus-host membrane fusion[3]. Despite these functional constraints, the receptor-binding site (RBS) of human H3N2 virus is also under strong positive selection pressure from herd immunity, since it partially overlaps with major antigenic sites A, B, and D[4]. Therefore, the HA RBS of human H3N2 virus has evolved over time, resulting in antigenic drift

[1]Department of Biochemistry, University of Illinois Urbana-Champaign, Urbana, IL 61801, USA. [2]HKU-Pasteur Research Pole, School of Public Health, Li Ka Shing Faculty of Medicine, The University of Hong Kong, Hong Kong SAR, China. [3]Department of Molecular Medicine, The Scripps Research Institute, La Jolla, CA 92037, USA. [4]Department of Immunology and Microbiology, The Scripps Research Institute, La Jolla, CA 92037, USA. [5]Center for Biophysics and Quantitative Biology, University of Illinois Urbana-Champaign, Urbana, IL 61801, USA. [6]The Jockey Club School of Public Health and Primary Care, Faculty of Medicine, The Chinese University of Hong Kong, Hong Kong SAR, China. [7]Li Ka Shing Institute of Health Sciences, Faculty of Medicine, The Chinese University of Hong Kong, Hong Kong SAR, China. [8]Department of Cell Biology and Infection, Institut Pasteur, Paris, Cedex 75015, France. [9]Centre for Immunology and Infection, Hong Kong Science Park, Hong Kong SAR, China. [10]Carl R. Woese Institute for Genomic Biology, University of Illinois Urbana-Champaign, Urbana, IL 61801, USA. [11]S.H. Ho Research Centre for Infectious Diseases, The Chinese University of Hong Kong, Hong Kong SAR, China. [12]School of Biomedical Sciences, The Chinese University of Hong Kong, Hong Kong SAR, China. [13]Carle Illinois College of Medicine, University of Illinois Urbana-Champaign, Urbana, IL 61801, USA. [14]These authors contributed equally: Ruipeng Lei, Weiwen Liang, Wenhao O. Ouyang, Andrea Hernandez Garcia ✉e-mail: kapunmok@cuhk.edu.hk; nicwu@illinois.edu

as well as changes in receptor binding mode and specificity[2,5–13]. When H3N2 virus first emerged in human, its HA receptor specificity switched from α2,3-linked sialylated glycan (avian-type receptor) to α2,6-linked sialylated glycan (human-type receptor)[14,15]. In the past two decades, human H3N2 virus has further evolved its HA receptor specificity towards a subset of α2,6-linked sialylated glycans that are branched and have extended poly-N-acetyl-lactosamine chains[12]. This evolution of receptor specificity is attributed to HA RBS mutations that form additional interactions beyond the terminal sialic acid (Sia-1)[2,16]. Many of these mutations are also associated with antigenic drift[2,17].

When human H3N2 virus replicates in chicken eggs for egg-grown seasonal influenza vaccine production, its receptor specificity often adapts to avian-type receptor by acquiring egg-adaptive mutations[18–20]. Certain egg-adaptive mutations can alter HA antigenicity and hence hamper vaccine efficacy[20–23], as exemplified by L194P mutation[23–25]. While L194P was a common egg-adaptive mutation in egg-grown H3N2 vaccine strains prior to 2020, it is rarely observed in recent years[26]. Consistently, our previous study has shown that L194P is incompatible with recent human H3N2 strains[26]. In other words, L194P imposes a huge replication fitness cost to human H3N2 strains from recent years but not before 2020. This observation demonstrates that epistasis exists between L194P and recently emerged natural mutations in human H3N2 HA. It also suggests that recent evolution of human H3N2 virus involves structural changes in the HA RBS. Given that most seasonal influenza vaccines are still produced in eggs and different egg-adaptive mutations have different antigenic effects[27], it is critical to understand of how natural mutations in human H3N2 HA influence the preference of egg-adaptive mutations.

Using X-ray crystallography, we demonstrated that the receptor binding mode of recent human H3N2 HA has evolved further. Structural analysis indicated that this evolution was coordinated by two natural mutations, G186D and D190N. Phylogenetic analysis revealed that G186D and D190N have coevolved in two different human H3N2 clades. Based on glycan binding analysis, this coevolution of G186D and D190N was due to epistasis. Specifically, both G186D and D190N were required to maintain the binding avidity to α2,6-linked sialylated glycans, whereas either of the mutations alone weakened this binding avidity. Our data also suggested that both G186D and D190N alters the HA antigenicity. Furthermore, using combinatorial mutagenesis and next-generation sequencing, we identified five spatially distant mutations in the HA RBS, including G186D and D190N, that conferred recent H3N2 strains with L194P incompatibility.

## Results
### Evolution of receptor binding mode in recent human H3N2 HA
Since January 2020, human H3N2 clades 3C.2a1b.1a and 3C.2a1b.2a2 have significantly expanded[26,28]. These two human H3N2 clades share the HA mutation D190N, which locates at the center of the RBS[26]. This observation led us to hypothesize that the receptor binding mode has evolved in recent human H3N2 HA. We thus determined the crystal structure of the HA from a clade 3C.2a1b.1a strain A/Victoria/22/2020 (Vic20, GISAID accession number: EPI1721444) in complex with a human-type receptor analog LSTc (NeuAcα2-6Galβ1-4GlcNAcβ1-3Galβ1-4Glc) to 2.16 Å resolution (Supplementary Table 1 and Supplementary Fig. 1).

To examine if Vic20 HA showed an evolved receptor binding mode, we compared our crystal structure with two previously determined crystal structures, namely HA from H3N2 A/Hong Kong/1/1968 (HK68) in complex with a human-type receptor analog 6′SLNLN (NeuAcα2-6Galβ1-4GlcNAcβ1-3Galβ1-4GlcNAc) and HA from H3N2 A/Ecuador/1374/2016 (Ecu16) in complex with 6′SLNLN (Fig. 1)[2,29]. The only difference between LSTc and 6′SLNLN is the C2 substituent in the fifth glycan moiety, which pointed away from and did not interact with

HA. Vic20 HA-bound LSTc exhibited an energetically favorable folded-back conformation[30], which is typically adopted by HA-bound human-type receptor analogs (Fig. 1a, b)[31]. Nevertheless, the human-type receptor analog in Vic20 was tilted upward starting from GlcNAc-3, as compared to that observed in HK68 and Ecu16 HAs (Fig. 1a). Moreover, GlcNAc-3 of the human-type receptor analog in Vic20 HA was closer to the 190-helix (Fig. 1b).

Ecu16 HA and Vic20 HA differed by three mutations in the RBS, namely T135K, G186D, and D190N (Fig. 1d-e). Since position 135 only interacted with the Sia-1 moiety of the human-type receptor analog, which had an almost identical positioning in Ecu16 HA and Vic20 HA, T135K did not seem to have a major role in the evolution of receptor binding mode. Conversely, the evolved receptor binding mode in Vic20 was largely determined by G186D and D190N. D190 in Ecu16 HA formed two water-mediated H-bonds with Sia-1 and GlcNAc-3 of the human-type receptor analog, respectively (Fig. 1d), whereas N190 in Vic20 HA directly H-bonded with GlcNAc-3 of the human-type receptor analog (Fig. 1e). This difference was driven by G186D in Vic20 HA, which stabilized the side chain conformation of N190 via an H-bond and pushed it towards GlcNAc-3 (Fig. 1e). Meanwhile, GlcNAc-3 of the human-type receptor analog was pulled towards the 190-helix in Vic20 HA by directly interacting with N190. While the receptor binding mode of human H3N2 HA has evolved throughout the past decades[2,8,11,24], our structural analysis here demonstrates that this evolution continues in recent human H3N2 HA.

### Recent human H3N2 HA has an expanded receptor-binding site
Our previous study has shown that mutation at residue 186 can influence the height of the RBS[23]. We postulated that the height of its RBS has been altered with the presence of G186D mutation in Vic20 HA. The height of the HA RBS was analyzed by measuring the distance between the Cα of residue 190 (Cα$_{190}$) and the phenolic oxygen of residue 98 (OH$_{98}$) in the apo form[11,23]. The Cα$_{190}$–OH$_{98}$ distance in HK68 HA with S186/E190 was around 8.9–9.2 Å, while it decreased to 8.5–8.9 Å in Fin04, HK05, Bris07, Vic11, and Mich14 HAs, all of which carry G186/D190 (Fig. 1f, g). Interestingly, the Cα$_{190}$–OH$_{98}$ distance went back up to 9.4 Å, which was even larger than that in HK68 HA. Similarly, Vic20 and HK68 HAs had larger Cα$_{186}$–Cα$_{228}$ distance, which is another proxy for the height of the HA RBS[11], than Fin04, HK05, Bris07, Vic11, and Mich14 HAs. This difference is unlikely to be an artifact of crystal packing, because the RBS is not involved in crystal packing interface of these HA structures. Additionally, the HA apo structures of Fin04, HK05, Bris07, Vic11, Mich14, and Vic20 all belong to the space group H 3 2 and have the same crystal packing. Of note, although the height of the HA RBS was similar between HK68 and Vic20 HAs, their receptor binding modes were markedly different (Fig. 1a–c, e). While the large height of the HK68 HA RBS helped prevent a steric clash between E190 and Sia-1 (Fig. 1c)[11], that of Vic20 HA RBS promoted the H-bond formation between N190 and the amide nitrogen of GlcNAc-3 by shortening their distance (Fig. 1e). These observations provide further mechanistic insights into the evolution of receptor binding mode in recent human H3N2 HA.

### Coevolution of G186D and D190N in recent human H3N2 clades
Sequence and phylogenetic analysis showed that although G186 and D190 have dominated human H3N2 virus in the past 20 years, D186 and N190 are prevalent in recent human H3N2 strains (Fig. 2a, b)[26,28,29]. Interestingly, mutations G186D and D190N appeared to coevolve, since human H3N2 strains that contained either G186D or D190N but not both were rarely observed (Fig. 2a). Besides, G186D and D190N co-occurred in two phylogenetically distinct clades, namely 3C.2a1b.1a and 3C.2a1b.2a2. Such coevolution was consistent with the physical interaction between D186 and N190 (Fig. 1e). Specifically, the H-bond between D186 and N190 would be lost due to a missing H-bond

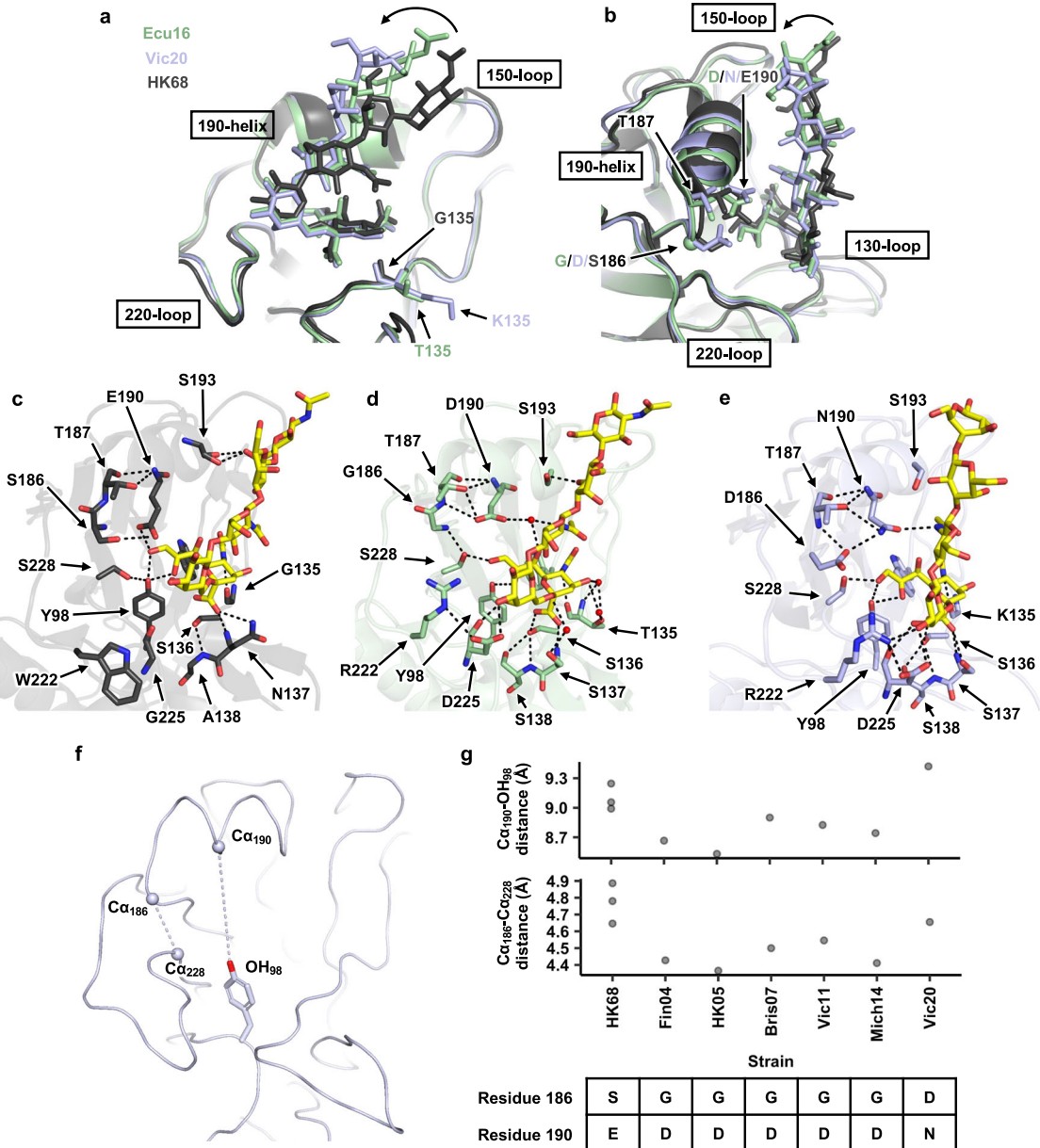

**Fig. 1 | Structural evolution of human H3N2 HA RBS. a, b** Structures of HK68 HA in complex with 6'SLNLN (gray, PDB 6TZB)[29], Ecu16 HA in complex with 6'SLNLN (green, PDB 8TJA)[2], and Vic20 HA in complex with LSTc (blue, PDB 8FAW from this study) are aligned based on the receptor-binding subdomain (HA1 residues 117-265)[36]. **a** Front and **b** side views are shown. Representative differences in receptor conformations are indicated by the curved arrow. **c** Interactions between HK68 HA and 6'SLNLN are shown. **d** Interactions between Ecu16 HA and 6'SLNLN are shown. **e** Interactions between Vic20 HA and LSTc are shown. Hydrogen bonds are represented by black dashed lines. Water molecules are shown as red spheres. Human-type receptor analogs (6'SLNLN and LSTc) are shown as yellow sticks. **f** The height of the RBS in HA apo form is measured by the distances between the Cα of residues 186 (Cα$_{186}$) and the Cα of residues 228 (Cα$_{228}$), as well as between the Cα of residues 190 (Cα$_{190}$) and the phenolic oxygen of residue 98 (OH$_{98}$). **g** Distances of Cα$_{190}$–OH$_{98}$ and Cα$_{186}$–Cα$_{228}$ in different human H3N2 HAs were measured. HK68: PDB 4FNK[59], Bris07: PDB 6AOQ[24], Fin04: PDB 2YP2[8], HK05: PDB 2YP7[8], Vic11: PDB 4O5N[60], Mich14: PDB 6BKP[11], Vic20 is from this study (PDB 8FAQ). Amino acid sequences at residues 186 and 190 on HA of different strains are shown. **f, g** Of note, the HA apo structure of Ecu16 was not available, and thus not included in this analysis. Raw data are provided as a Source Data file.

acceptor if residue 186 was a Gly (G186/N190), or a missing H-bond donor if residue 190 was an Asp (D186/D190). Moreover, if both residues 186 and 190 were Asp (D186/D190), there would be an unfavorable electrostatic repulsion between them. As described above, the H-bond between D186 and N190 stabilized the side chain conformation of N190, which in turn promoted the direct interaction between N190 and GlcNAc-3, and hence the evolution of receptor binding mode in recent human H3N2 HA (Fig. 1e). Therefore, the coevolution of G186D and D190N is likely related to their synergistic effect in receptor binding.

## Receptor binding and thermostability analyses reveal epistasis between G186D and D190N

To examine the functional importance of the interaction between D186 and N190, we recombinantly expressed wild type (WT) and three mutants of Vic20 HA, namely D186G, N190D, and double mutant D186G/N190D. Glycan array analysis showed that WT Vic20 HA bound strongly to extended biantennary α2,6-linked sialylated glycans (Fig. 3a, Supplementary Data 1). This observation is consistent with other human H3N2 HAs in the past 15 years[6,12]. While the double mutant D186G/N190D showed similar glycan array binding profile as WT, such

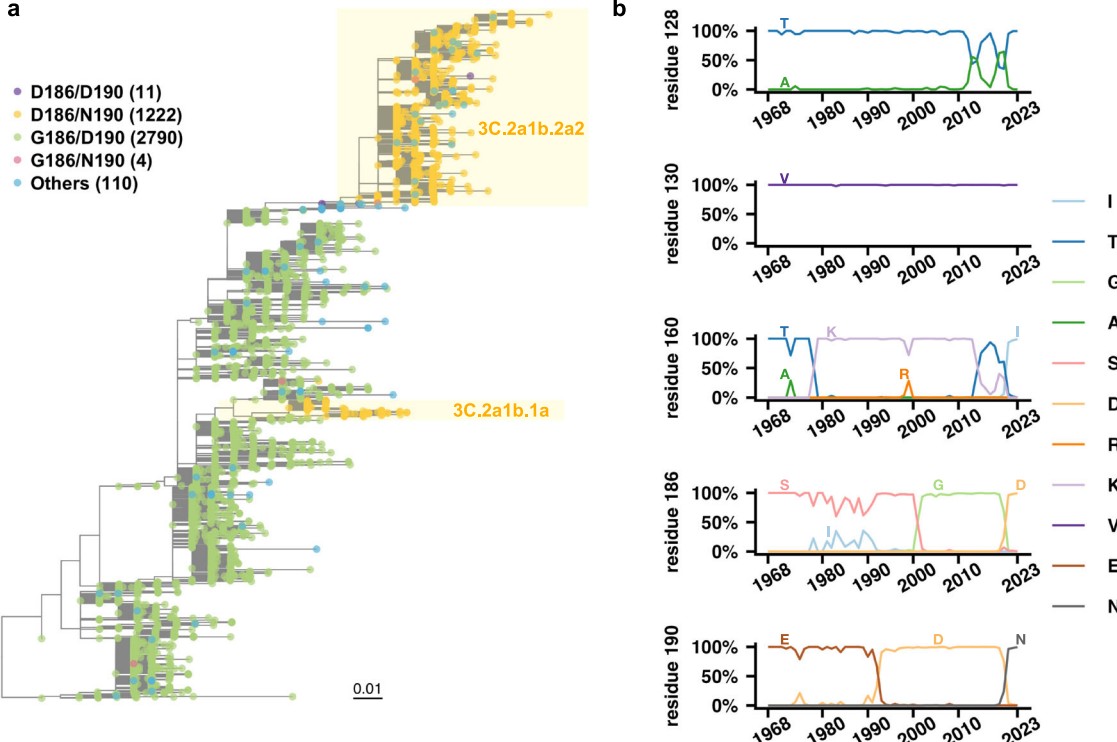

**Fig. 2 | Coevolution of G186D and D190N in the HA RBS of recent human H3N2 strains. a** A rooted phylogenetic tree was built on the protein sequences of receptor-binding subdomain (HA1 residues 117–265)[36] from human H3N2 strains that were isolated between January 2017 and June 2023. Scale bar refers to a phylogenetic distance of 0.01 amino acid mutations per site. Amino acid variants at residues 186 and 190 are color coded. **b** The natural occurrence frequencies of major amino acid variants at residues 128, 130, 160, 186, and 190 in human H3N2 HA over time are shown. Only variants that have reached at least 20% in any year are plotted. Raw data are provided as a Source Data file.

binding profile was weakened by N190D and almost abolished by D186G (Fig. 3a). Consistently, enzyme-linked immunosorbent assay (ELISA) showed that both WT and double mutant D186G/N190D, to a greater extent, bound strongly to biantennary α2,6-linked sialylated glycan 6′SLN$_3$-N, whereas this binding was weakened and abolished by single mutants N190D and D186G, respectively (Fig. 3b–f). A similar observation was made with linear α2,6-linked sialylated glycan 6′SLN$_3$-L, albeit with much weaker binding signal (Supplementary Fig. 2a–e). We also found that the thermostability effects of N190D and D186G were non-additive (Fig. 4a). While D186G and N190D decreased the melting temperature ($T_m$) of Vic20 HA by around 4 °C and 2 °C, respectively, the double mutant D186G/N190D decreased the $T_m$ of Vic20 HA by 3 °C rather than 6 °C as expected from additive interaction. In other words, although N190D had a destabilizing effect on WT Vic20 HA, it had a stabilizing effect when introduced on top of D186G. Altogether, these results imply that the coevolution of G186D and D190N can be attributed to epistasis, which is manifested in receptor binding avidity and protein thermostability.

## Sequence variation at residues 186 and 190 impacts HA antigenicity

Both residues 186 and 190 belong to major antigenic site B[4]. To investigate if G186D and D190N contribute to HA antigenic drift, we introduced D186G, N190D, and the double mutant D186G/N190D into the HA of H3N2 A/Italy/11871/2020 (Italy20, GISAID accession number: EPI1735610), which was also from clade 3C.2a1b.1a and shared >99% HA sequence identity with Vic20 HA (Supplementary Table 2). Although D186G and N190D were both deleterious for receptor binding, all Italy20 HA mutants could be rescued as 6:2 reassortants on the H1N1 A/Puerto Rico/8/1934 (PR8) backbone and grown to high viral titers in hMDCK cells (Supplementary Fig. 3). This observation was not

unexpected because reduction in HA receptor binding avidity does not necessarily diminish in vitro virus growth as shown in MDCK-SIAT1 cells or MDCK cells[32,33], which are the parental cell lines of hMDCK cells[34]. Subsequently, we obtained antisera from mice that were immunized intraperitoneally with adjuvanted Italy20 WT virus. The neutralizing activities of these antisera were tested against Italy20 WT, D186G, N190D, and the double mutant D186G/N190D using a microneutralization (MN) assay. While antisera from WT-immunized mice showed 50% microneutralization (MN$_{50}$) titers ranging from 1:80 to 1:640 against both WT and single mutant D186G, their MN$_{50}$ titers decreased significantly to a range of 1:20 to 1:160 against single mutant N190D and double mutant D186G/N190D ($P < 0.05$, paired Wilcoxon signed-rank test) (Fig. 4b). These results suggest that residue 190 is an antigenic determinant of Italy20 HA.

We further measured the neutralizing activities of plasma samples from human who had not been infected or vaccinated by recent H3N2 strains with D186/N190[35]. As expected, these plasma samples had significantly higher MN$_{50}$ titers against an older H3N2 strain in clade 3C.2a, namely A/Singapore/INFIMH-16-0019/2016 (Sing16), than Italy20 WT ($P < 0.05$, paired Wilcoxon signed-rank test) (Fig. 4c). Specifically, the average MN$_{50}$ titer decreased from above 1:160 against Sing16 to below 1:20 against Italy20 WT. Such decrease in MN$_{50}$ titers could be partly compensated to around 1:80 by D186G ($P < 0.05$, paired Wilcoxon signed-rank test, Fig. 4c). This observation suggests that G186D is involved in the antigenic drift of human H3N2 virus.

## Mapping sequence determinants for L194P incompatibility in recent H3N2 strains

Before 2020, L194P is a common egg-adaptive mutation that alters the HA antigenicity of egg-grown H3N2 vaccine strains[24,26]. Nevertheless, L194P is rarely observed in recent egg-grown H3N2 vaccine strains, due

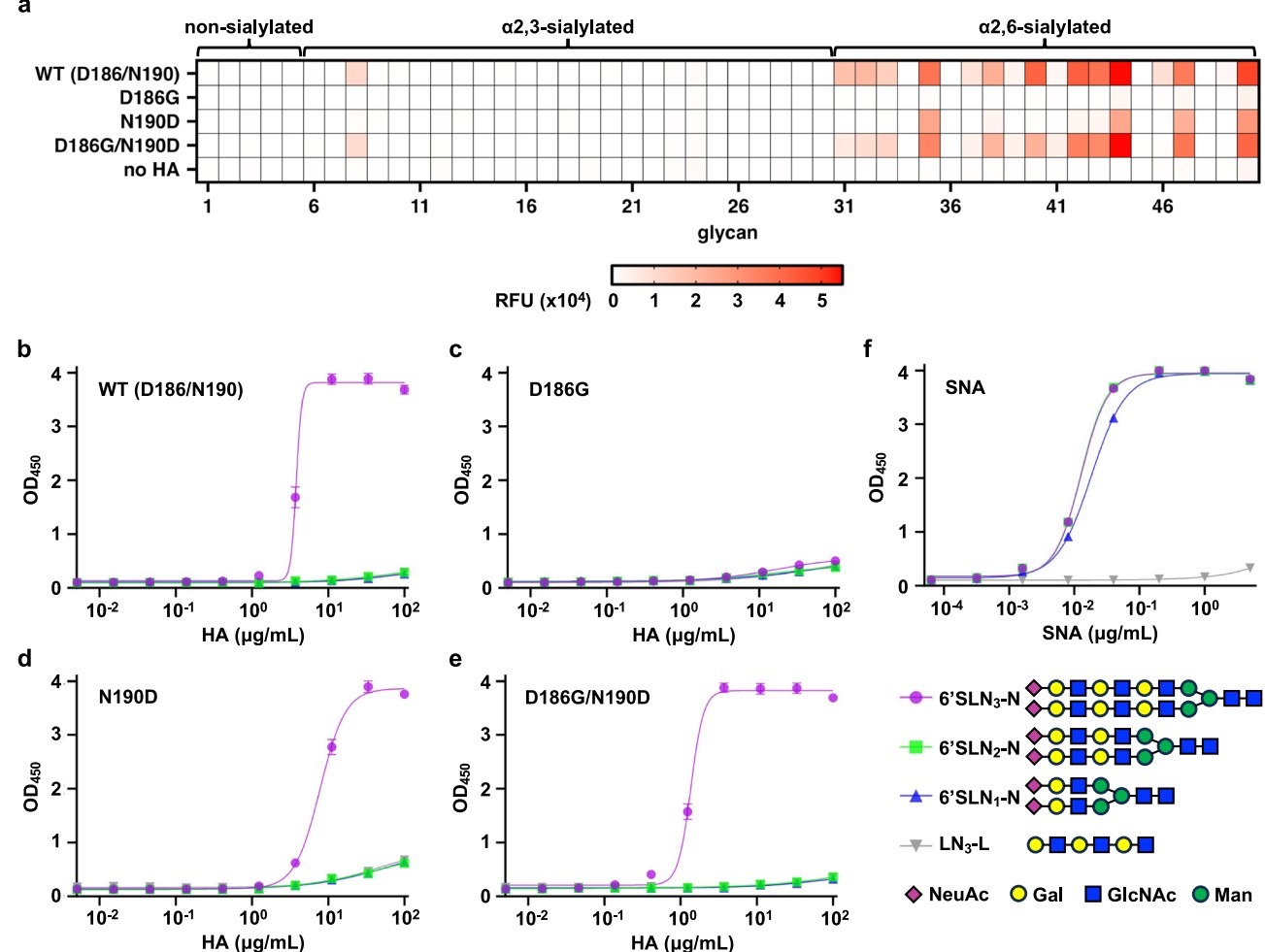

**Fig. 3 | Glycan binding analysis of Vic20 HA WT and mutants. a** Glycan array analysis of recombinant Vic20 HA proteins (WT and mutants, *y*-axis). Binding activity was measured by relative fluorescence unit (RFU) and represented by a heatmap. The corresponding glycan for each glycan ID (*x*-axis) is shown in Supplementary Data 1. **b–f** Binding avidities of **b–e** recombinant Vic20 HAs (WT and mutants) as well as **f** sambucus nigra agglutinin (SNA, positive control) to the indicated glycans were measured by ELISA. LN₃-L was used as a negative control. The means of optical density 450 nm (OD₄₅₀) from three independent experiments are shown with SD indicated by the error bars. -N: N-linked, -L: linear. Glycan diagrams are drawn according to the Symbol Nomenclature for Glycans recommended by the NLM[61,62]. Raw data are provided as a Source Data file.

to epistatic interactions with natural mutations that emerged during HA RBS evolution[26]. Such epistatic interactions lead to an incompatibility between L194P and recent human H3N2 strains. Previously, we have identified that two natural mutations K160T and D190N are incompatible with L194P[26]. However, reverting T160/N190 in recent human H3N2 strains to K160/D190 fails to restore the L194P compatibility[26], indicating that additional natural mutation may also play a role in regulating the L194P compatibility.

Here, we aimed to systematically identify natural mutations in human H3N2 strains that are incompatible with L194P. The amino acid sequences of HA receptor-binding subdomain (HA1 residues 117–265)[36] were compared between Italy20 and Sing16, which is an ancestral strain of clade 3C.2a and can adapt to eggs by mutations T160K, L194P, and D225G (GISAID accession number: EPI1082230). In addition to L194P, there were ten mutations in the HA receptor-binding subdomain between Italy20 and the egg-adapted Sing16, namely A128T, I130V, K135T, S138A, T160K, D186G, N190D, S193F, P198S, and D225G (Fig. 5a). We then probed the compatibility between L194P and each of the possible combinations of the above mutations ($2^{10} = 1024$ variants) using combinatorial mutagenesis and next-generation sequencing[11,29,37]. Briefly, a plasmid mutant library containing all possible combinations of the above mutations was constructed in the genetic background of Italy20 HA with L194P (Italy20-L194P). A virus mutant library was then generated from the plasmid mutant library using the influenza reverse genetics system[38], and passaged once in MDCK-SIAT1 cells. Subsequently, next-generation sequencing was performed to monitor the enrichment of individual variants after passaging. Three variants showed strong enrichment after passaging, indicating that they could restore the L194P compatibility (Fig. 5b). These three variants all contained mutations A128T, I130V, K135T, T160K, D186G, N190D, and S193F. Consistently, our validation experiment showed that a combination of these seven mutations (named as Mut7 in Fig. 5c) restored the viral titer of Italy20-L194P to WT-like level in hMDCK cells.

Next, we further investigated the importance of each mutation in Mut7. A series of recombinant mutants were generated by introducing only six out of the seven mutations in Mut7 into Italy20-L194P (e.g., Mut7 without A128T). Consistent with our previous study[26], our mutagenesis results showed that T160 and N190 contributed to the L194P incompatibility in Italy20, since Mut7 without T160K or N190D could not compensate the fitness defect of Italy20-L194P (Fig. 5c). Additionally, A128, I130, and D186 contributed to the L194P incompatibility in Italy20. Of note, I130 is a rare variant in human H3N2 HA (Fig. 2b).

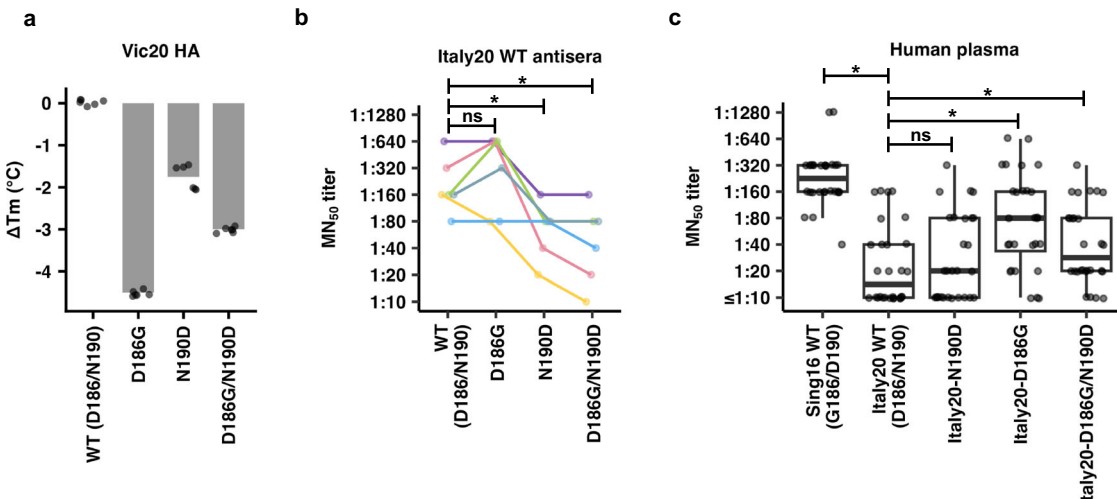

**Fig. 4 | Thermostability and antigenic effects of G186D and D190N mutations.**
**a** Recombinant Vic20 HAs (WT and mutants) were analyzed by thermal shift assay. The differences in melting temperature ($\Delta T_m$) between WT and mutants are shown. Each datapoint represents one experimental replicate. Each bar represents the mean of six independent replicates. **(b)** A group of 6-week-old female BALB/c mice ($n = 6$) were immunized intraperitoneally with 10,000 PFU of recombinant influenza virus with HA and NA from Italy20 and the six internal segments from PR8. On day 21 post-immunization, a booster was performed with the same amount of adjuvanted virus. Antisera were collected on day 21 post-boosting. 50% micro-neutralization ($MN_{50}$) titer of the antisera against Italy20 WT and mutants was measured by a microneutralization (MN) assay. Each color represents the anti-serum from one mouse. **c** $MN_{50}$ titer of human plasma ($n = 28$) against an older human H3N2 strain Sing16 WT as well as Italy20 WT and mutants were measured by MN assay. Data are presented as a boxplot. The middle horizontal line represents the median. The lower and upper hinges represent the first and third quartiles, respectively. The upper whisker extends to the highest data point within a 1.5× inter-quartile range (IQR) of the third quartile, whereas the lower whisker extends to the lowest data point within a 1.5× IQR of the first quartile. Each data point represents one plasma sample. **b, c** The median $MN_{50}$ titers were compared between Italy20 WT and different mutants using two-sided paired Wilcoxon signed-rank test without adjustment for multiple comparisons. $P$-value was computed between groups, where "*" indicates statistically significant difference ($P < 0.05$), whereas "ns" indicates non-significant. **b** The $P$-values for WT vs D186G, WT vs N190D, and WT vs D186G/N190D are 0.14, 0.04, and 0.03, respectively. **c** The $P$-values for Italy20 WT vs Sing16 WT, Italy20 WT vs Italy20 N190D, Italy20 WT vs Italy20 D186G, and Italy20 WT vs Italy20 D186G/N190D are 8e−6, 0.22, 7e−5, and 0.002, respectively. Raw data are provided as a Source Data file.

As mentioned above, 3C.2a1b.1a and 3C.2a1b.2a2 are the two recent human H3N2 clades with G186D and D190N. Unlike clade 3C.2a1b.1a, which has an N-glycosylation site at residue 158, clade 3C.2a1b.2a2 has lost this N-glycosylation site via mutation T160I (Supplementary Fig. 4a)[6]. Given that T160K helped restore L194P compatibility in Italy20, we aimed to further test if T160I could do the same by replacing T160K in Mut7 by T160I. Our rescue experiment showed that Mut7 with T160I instead of T160K failed to compensate the fitness defect of Italy20-L194P (Supplementary Fig. 4b), despite both mutations led to a loss of an N-glycosylation site at residue 158. This observation suggests that I160 may also contribute to the L194P incompatibility in clade 3C.2a1b.2a2 through a mechanism unrelated to N-glycosylation. In summary, our results demonstrate that the fitness effect of L194P mutation is influenced by multiple mutations across the 130-loop, 150-loop, and 190-helix of RBS (Fig. 5d), substantiating that epistasis is pervasive among RBS mutations[11,29,32].

## Discussion

The HA receptor binding mode of human H3N2 virus has been evolving since its emergence in 1968[2,8,11,24,29]. In our present study, we demonstrated that the receptor binding mode of recent human H3N2 HA has evolved further, mainly due to mutations G186D/D190N, which pulled the GlcNAc-3 of human-type receptor closer to the 190-helix. Our glycan binding analysis revealed that G186D and D190N interacted epistatically, where either G186D or D190N alone diminished receptor binding. This result indicates that circulating human H3N2 virus can cross fitness valleys to explore evolutionary trajectories that are otherwise constrained by epistasis, through rapidly, or even simultaneously, acquiring at least two mutations.

HA residue 190 is located at the center of the RBS and three major amino acid variants have been observed in human H3N2 viruses. It was a Glu when human H3N2 virus first emerged in human in 1968, mutated

to Asp during the 1992–1993 influenza season, and mutated again to Asn in two recent clades 3C.2a1b.1a and 3C.2a1b.2a2 (Fig. 2a, b). As clade 3C.2a1b.2a2 was the predominant clade during the 2023–2024 influenza season, Asn has reached fixation[28]. E190 in early human H3N2 virus H-bonds with Sia-1[29,39]. D190 in human H3N2 virus from 2000s to 2010s forms water-mediated H-bonds with both Sia-1 and GlcNAc-3[2]. As shown in this study, N190 in recent human H3N2 virus directly H-bonds with GlcNAc-3 without interacting with Sia-1. Thus, the evolution of residue 190 is consistent with the notion that human H3N2 HA RBS has evolved additional contacts along the length of the receptor glycan chain rather than the terminal Sia-1[2].

Previous studies have shown that D190N occurs in egg-grown but not circulating human H3N2 strains from clade 3C.3a[26,40]. It is unclear how D190N acts as an egg-adaptive mutation and at the same time emerges in human H3N2 virus with strong binding preference for human-type receptor. One possibility is that D190N, albeit being an egg-adaptive mutation, does not facilitate binding to avian-type receptor. Another mutually non-exclusive possibility is that the effect of D190N on receptor specificity is strain-dependent. We postulate that the latter is likely, given that most egg-adaptive mutations are known to increase binding avidity to avian-type receptor[18,21,23,24,41] and epistatic interactions are prevalence in the HA RBS[11,29,32]. Nevertheless, this perplexing observation of D190N being both an egg-adaptive mutation and a natural mutation in circulating strains highlights that the sequence determinants of HA receptor specificity remain to be fully characterized.

Another notable finding in our study is that the natural mutations T128A and V130I are incompatible with L194P. These are examples of long-range epistasis since residues 128 and 130 are spatially distal to residue 194. Our previous study has demonstrated that L194P incompatibility can be conferred by an increased height of the HA RBS[23]. Consistently, Vic20 HA, which has a higher RBS height compared to

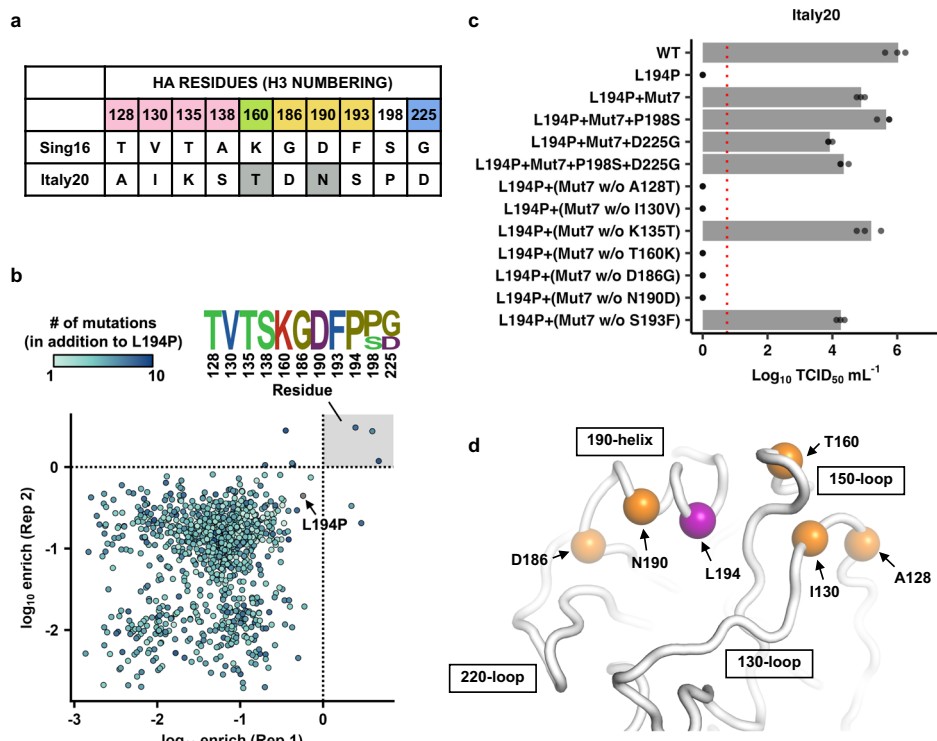

**Fig. 5 | Mapping natural mutations that influence compatibility with the egg-adaptive mutation L194P. a** Differences in the amino acid sequences of receptor-binding subdomain (HA1 residues 117–265)[36] between the egg-adapted Sing16 (with egg-adaptive mutations T160K, L194P, and D225G) and Italy20 are shown. Residues in the four structural elements of the RBS, namely 130-loop, 150-loop, 190-helix and 220-loop are highlighted in pink, green, yellow and blue, respectively. T160 and N190, which were previously demonstrated to be incompatible with L194P[26], are highlighted in gray. **b** The relative replication fitness effects of all possible combinations of A128T, I130V, K135T, S138A, T160K, D186G, N190D, S193F, P198S, and D225G on Italy20 with L194P mutation were assessed by combinatorial mutagenesis and next-generation sequencing. Relationship of the $\log_{10}$ enrichment values between two independent replicates is shown as a scatterplot. Each data point

represents one variant and is colored based on the number of mutations. Sequences of variants with a $\log_{10}$ enrichment of >0 in both replicates are represented by a sequence logo. **c** The replication fitness effects of Italy20 with different combinations of mutations were examined by virus rescue experiments. Mut7 represents a combination of seven mutations in 130-loop, 150-loop, and 190-helix (A128T, I130V, K135T, T160K, D186G, N190D, and S193F). Viral titers were measured by $TCID_{50}$[48,49]. Each data point represents the viral titer of an independent replicate ($n = 3$). The mean is represented by the bar. The dashed red line represents the lower detection limit. **d** Structure of Vic20 HA RBS (PDB 8FAQ) is shown. Amino acid residues incompatible with the L194P mutation are highlighted in orange. The location of L194 is highlighted in purple. Raw data are provided as a Source Data file.

that of older strains, is incompatible with L194P[26]. Future studies are needed to investigate whether T128A and V130I increase the height of the RBS, or if they confer L194P incompatibility through a previously unknown mechanism. Regardless, our observation here implies that the natural evolution of human H3N2 HA RBS can be shaped by long-range epistasis between distant residues. Due to the public health relevance of HA RBS evolution, continued studies of its functional constraints and the underlying biophysical basis are warranted.

# Methods
## Ethical statement
The study design and conduct complied with all relevant regulations regarding the use of human study participants and was conducted in accordance to the criteria set by the Declaration of Helsinki. Protocol for collecting human plasma samples was approved by the Joint Chinese University of Hong Kong-New Territories East Cluster (Ref no: 2020.229) Clinical Research Ethics Committee. All animal procedures were carried out in accordance with institutionally approved protocols of The University of Hong Kong (Approval number: 5598-20).

## Phylogenetic tree analysis
Protein sequences of HA from human H3N2 strains isolated between January 2017 and June 2023 were downloaded from the Global Initiative for Sharing Avian Influenza Data (GISAID; http://gisaid.org)[42]. The sequences of receptor-binding subdomain (HA1 residues 117–265, H3

numbering)[36] were aligned using MAFFT v7 with default parameters[43]. Duplicate sequences and sequences from strains that had an ambiguous passaging history or were passaged in eggs were removed as described[44]. A phylogenetic tree was constructed using IQ-TREE2[45] with default parameters and visualized using the ggtree package in R[46].

## Cell culture
Human embryonic kidney (HEK) 293 T cells (ATCC, cat no.CRL-3216), humanized Madin-Darby canine kidney (hMDCK) cells (from Yoshihiro Kawaoka)[34], and MDCK-SIAT1 cells (Sigma-Aldrich cat no.05071502) were used in this study. These cell lines were maintained in Dulbecco's modified Eagle's medium (DMEM, Gibco, cat no. 11965092) supplemented with 10% (v/v) fetal bovine serum (FBS, Gibco, cat no. 10082147), 25 mM HEPES (Gibco, cat no. 15630080), and 100 U mL$^{-1}$ penicillin-streptomycin (PS, Gibco, cat no. 15140122) at 37 °C, 5% $CO_2$ and 95% humidity. Sf9 cells (*Spodoptera frugiperda* ovarian cells, female, ATCC, cat no. CRL-1711) were maintained in Sf-900 II SFM medium (Thermo Fisher Scientific, cat no. 10902088) at 37 °C. Expi293F cells (Gibco, cat no. A14527) were grown and maintained in Expi293 Expression Medium (Gibco, cat no. A1435101) at 37 °C, 8% $CO_2$, and 95% humidity with shaking at 125 rpm according to the manufacturer's instructions.

## Virus rescue experiments
Influenza H3N2 viruses were generated by the influenza eight-plasmid reverse genetics system using H1N1 A/Puerto Rico/8/34 (PR8) as a

backbone[38,47]. The HA and neuraminidase (NA) genes of Italy20 (GISAID accession numbers: EPI1735610 and EPI1735609, respectively) were synthesized by Sangon Biotech, and cloned into the pHW2000 vector as previously described[26]. Briefly, Italy20 HA ectodomain was flanked by the non-coding regions, N-terminal secretion signal, C-terminal transmembrane domain, and cytoplasmic tail from PR8. For NA, the non-coding regions were from PR8, whereas the coding region was from Italy20. Mutations were introduced by polymerase chain reaction (PCR) using PrimeSTAR MAX DNA polymerase (Takara Bio, cat no. R045B) according to manufacturer's instructions, with primers from Integrated DNA Technologies. Recombinant 6:2 viruses with the six internal genes (PB2, PB1, PA, NP, M, and NS) from PR8 were rescued as previously described[26]. In brief, transfection was performed in a co-culture of HEK 293 T and hMDCK cells (ratio of 6:1) at 70% confluence in a 6-well plate. For each recombinant virus, 16 µL of TransIT-LT1 (Mirus Bio, cat no. MIR 2304) and 1 µg of each of the eight plasmids encoding individual influenza virus segments were mixed and incubated for 20 min at room temperature before adding to the cells. At 6 h post-transfection, medium was replaced with 1 mL of DMEM. At 24 h post-transfection, another 1 mL of DMEM supplemented with 100 U mL$^{-1}$ PS, 25 mM HEPES and 1 µg mL$^{-1}$ tosylphenylalanyl chloromethyl ketone (TPCK)-trypsin (Sigma-Aldrich, cat no. T1426) was added. Supernatant was harvested at 72 h post-transfection, all of which was then inoculated into hMDCK cells at 90% confluence in a T75 flask with DMEM containing 1 µg mL$^{-1}$ TPCK-trypsin. Supernatant of the hMDCK cells was harvested when more than 70% of cells showed cytopathic effect. For each recombinant virus, three independent rescue experiments were performed. All rescue experiments included a positive control that contained wild type (WT) HA and a negative control that contained no HA. Each rescuable virus was further plaque purified. The identity of virus was confirmed by Sanger sequencing of the HA after viral RNA extraction using QIAamp Viral RNA Mini Kit (Qiagen, cat no. 52904), reverse transcription by ProtoScript II First Strand cDNA Synthesis Kit (New England Biolabs, cat no. E6560L), and amplified by PCR. Titer of the rescued virus was measured by TCID$_{50}$ assay in hMDCK cells as previously described[48,49]. Briefly, hMDCK cells were prepared in 96-well plate and cultured at 37 °C with 5% CO$_2$ until 90% confluence was obtained. Cells were washed with 1× phosphate-buffered saline (PBS, Gibco, cat no. 10010023) once and replenished with 50 µL DMEM supplemented with 25 mM HEPES and 100 U mL$^{-1}$ PS in each well. Virus stock was serially diluted at half-log increments and subsequently added to hMDCK cells in four replicate wells before incubating at 37 °C and 5% CO$_2$ for 1 h. Cell supernatants were then discarded and replaced with 150 µL of DMEM supplemented with 25 mM HEPES, 100 U mL$^{-1}$ PS, and 1 µg mL$^{-1}$ TPCK-trypsin in each well before incubating at 37 °C and 5% CO$_2$ for another 72 h. The TCID$_{50}$ titer was calculated using the Reed–Muench method.

## Passaging of recombinant influenza virus

hMDCK cells were washed with PBS twice and infected with viruses at a multiplicity of infection (MOI) of 0.001 in DMEM supplemented with 25 mM HEPES, 100 U mL$^{-1}$ PS, and 1 µg mL$^{-1}$ TPCK-trypsin. Viral supernatants were harvested at 24 h and 48 h post-infection and titered by TCID$_{50}$ assay using hMDCK cells as described above.

## Construction of mutant library

Mutant library was generated by a ligation strategy. pHW2000-Italy20 HA with L194P was used as the template for both insert and vector. Three PCRs tiling the entire insert region were performed using three sets of primers #1-F/R, #2-F/R, and #3-F/R (Supplementary Table 3). The resulting three PCR products were mixed together at equal molar ratio and amplified by PCR using primers #1-F and #3-R (Supplementary Table 3) to generate the final insert for the combinatorial mutant library. The vector was amplified using primers: 5′-CGT ACG TCT CAA GCA TCT ATT GGA CAA TAG TAA AAC-3′ and 5′-CGT ACG TCT CAC

CAA TTG AAG CTT TCA TTT TTA AAC-3′. All PCRs were performed using PrimeSTAR Max polymerase according to the manufacturer's instructions, followed by purification using the Monarch Gel Extraction Kit (New England Biolabs, cat no. T1020S). The final insert and the vector were digested by BsmBI-v2 (New England Biolabs, cat no. R0739S) and ligated using T4 DNA Ligase (New England Biolabs, cat no. M0202S). Ligation product was purified by the PureLink PCR Purification Kit (Thermo Fisher Scientific, cat no. K310001) and then transformed into MegaX DH10B T1R cells (Thermo Fisher Scientific, cat no. C640003). At least half a million colonies were collected. The plasmid mutant library was purified from the bacterial colonies using the PureLink HiPure Plasmid Midiprep Kit (Thermo Fisher Scientific, cat no. K210005). All primers were ordered from Integrated DNA Technologies.

## High-throughput fitness measurement of HA mutants

The plasmid HA mutant library and the other seven segments from H1N1 A/WSN/1933 at equal mass ratio were co-transfected into a co-culture of HEK 293 T/MDCK-SIAT1 cells (ratio of 6:1) at 60% confluence in a T75 flask (75 cm$^2$) using Lipofectamine 2000 (Thermo Fisher Scientifc, cat no. 11668019) according to the manufacturer's instructions. At 24 h post-transfection, cells were washed twice with PBS and the cell culture medium was replaced with OPTI-MEM medium (Gibco, cat no. 31985070) supplemented with 0.8 µg mL$^{-1}$ TPCK-trypsin. Supernatant containing the virus mutant library was harvested at 96 h post-transfection and titered by TCID$_{50}$ assay using MDCK-SIAT1 cells as described above. The rescued virus mutant library was stored at −80 °C until use.

For passaging the rescued virus mutant library, MDCK-SIAT1 cells in a T75 flask were washed twice with PBS and infected at an MOI of 0.02 in OPTI-MEM medium containing 0.8 µg mL$^{-1}$ TPCK-trypsin. Infected cells were washed twice with PBS at 2 h post-infection and resupplied with OPTI-MEM medium containing 0.8 µg mL$^{-1}$ TPCK-trypsin. At 24 h post-infection, supernatant containing the virus library was collected. Viral RNA was extracted using QIAamp Viral RNA Mini Kit. Purified viral RNA was reverse transcribed to cDNA using Superscript III reverse transcriptase (Thermo Fisher Scientific, cat no. 18080093), and then amplified using KOD Hot Start DNA polymerase (MilliporeSigma, cat no. 71086-3CN) according to the manufacturer's instruction using primers: 5′-CAC TCT TTC CCT ACA CGA CGC TCT TCC GAT CTA GTT GCC TCA TCC GGC ACA CTG-3′ and 5′-GAC TGG AGT TCA GAC GTG TGC TCT TCC GAT CTC AAT AGA TGC TTA TTC TGC TAG-3′. The plasmid mutant library was also amplified by PCR in the same manner. A second round of PCR was carried out to add the adapter sequence and index to the amplicons using primers 5′-AAT GAT ACG GCG ACC ACC GAG ATC TAC ACT CTT TCC CTA CAC GAC GCT-3′ and 5′-CAA GCA GAA GAC GGC ATA CGA GAT XXX XXX GTG ACT GGA GTT CAG ACG TGT GCT-3′ as described previously[37]. Nucleotides in primers annotated by "X" represented the index sequence for distinguishing PCR products derived from different samples. Finally, the PCR products were analyzed by next-generation sequencing using Illumina MiSeq PE250.

## Analysis of next-generating sequencing data

Next-generation sequencing data were obtained in FASTQ format. Forward and reverse reads of each paired-end read were merged by PEAR[50]. The merged reads were parsed by SeqIO module in BioPython[51]. Primer sequences were trimmed from the merged reads. Trimmed reads with a length that differed from the expected value were discarded. Subsequently, the trimmed reads were translated into amino acid sequences with sequencing error correction performed[52]. Mutations were called by comparing the translated reads to the amino acid sequence of Italy20 HA. Translated reads that did not contain the L194P mutation were removed from downstream analysis. The frequency ($F$) of a given mutant $i$ in a given sample $n$ was computed as

follows:

$$F_{i,n} = \frac{\text{readcount}_{i,n} + 1}{\sum_i (\text{readcount}_{i,n} + 1)} \quad (1)$$

A pseudocount of 1 was added to the read counts of each mutant to avoid division by zero in subsequent steps. The enrichment ($E$) of a given mutant $i$ in replicate $n$ of the post-passaged virus mutant library was then calculated using the following equation:

$$E_{i,n} = \frac{F_{i,n}}{F_{i,\text{plasmid mutant library}}} \quad (2)$$

## Crystallization and structural determination of HA protein

The Vic20 HA ectodomain, which contained HA1 residues 11–329 and HA2 residues 1–176, was fused to an N-terminal gp67 signal peptide and to a C-terminal biotinylation site, trimerization domain, and 6× His-tag. Recombinant bacmid DNA that carried the Vic20 HA ectodomain construct was generated using the Bac-to-Bac system (Thermo Fisher Scientific, cat no. 10359016). Baculovirus was generated by transfecting purified bacmid DNA into adherent Sf9 cells using Cellfectin (Thermo Fisher Scientific, cat no. 10362100) according to the manufacturer's instructions. The baculovirus was further amplified by passaging in adherent Sf9 cells at an MOI of 1. Recombinant HA ectodomain were expressed by infecting 1 L of suspension Sf9 cells at an MOI of 1. Infected Sf9 cells were incubated at 27 °C with shaking at 170 rpm for 72 h. HA0 was purified from the supernatant by Ni-NTA, buffer exchanged into 20 mM Tris-HCl, pH 8.0, with 100 mM NaCl, digested with trypsin (New England Biolabs, cat no. P8101S), and purified by size exclusion chromatography on a HiLoad 16/100 Superdex 200 column (Cytiva, cat no. 90100137). Crystallization screening was performed using the JCSG Core Suites I–IV (Rigaku) with HA at 7 mg mL$^{-1}$. Sitting drop for crystallization screening was set up by equal volume of precipitant and protein solution using the Crystal Gryphon (Art Robbins Instruments). Crystallization screens were incubated at 18 °C. Initial hits were further optimized using the sitting drop method at 18 °C, with 350 μL of reservoir solution and 1:1 ratio of precipitant and protein solution. Diffraction-quality crystals were obtained from 50% PEG-200 and 0.1 M citrate pH 5.5. The resulting crystals were vitrified and store in liquid nitrogen until data collection. To obtain the LSTc-bound structure, crystals were soaked for 3 h at 18 °C with the precipitant condition containing 15 mM of LSTc ligand prior to vitrification.

Data were collected at the advanced photon source (APS) at Argonne National Laboratory via the Life Science Collaborative Access Team (LS-CAT) at beamlines 21-ID-D for the Vic20 HA apo crystal structure, and 21-ID-F for the LSTc-bound Vic20 HA crystal structure. Initial diffraction data were indexed, integrated, and scaled using autoPROC[53]. The structure was solved by molecular replacement using Phaser-MR included in the Phenix suite[53], using as PDB 6BKP as the replacement model[11]. The structure was further refined using REFMAC5[54] and was manually built in COOT[55]. Ramachandran statistics were calculated using MolProbity[56].

## Thermal shift assay

Five μg of protein was mixed with 5× SYPRO orange (Thermo Fisher Scientific, cat no. S6650) in 20 mM Tris-HCl pH 8.0, 100 mM NaCl, and 10 mM CaCl$_2$ at a final volume of 25 μL. The sample mixture was then transferred into an optically-clear PCR tube (VWR). SYPRO orange fluorescence data in relative fluorescence unit (RFU) was collected from 10 °C to 95 °C using a CFX Connect Real-Time PCR Detection System (Bio-Rad). The temperature corresponding to the lowest point of the first derivative −d(RFU)/d$T$ was determined to be the melting temperature ($T_m$).

## Glycan array analysis

An N-terminal Igκ signal peptide, Vic20 HA ectodomain, which contained HA1 residues 11–329 and HA2 residues 1–176, a C-terminal biotinylation site, trimerization domain, and 6× His-tag were all cloned in-frame in a phCMV3 vector. Mutations D186G and/or N190D were introduced using PCR via site-directed mutagenesis. Expi293F cells were transfected with the plasmid encoding soluble HA with the Expifectamine 293 Transfection Kit (Gibco, cat no. A14526) according to the manufacturer's instructions. Six days post-transfection, the supernatant was collected by centrifugation of the cell suspension at 4500×$g$, 4 °C for 45 min. The supernatant was clarified using a polyethersulfone membrane with a 0.22 μm filter (Millipore). Soluble Vic20 HA was purified from supernatant using Ni-NTA, and further purified by size exclusion chromatography on a HiLoad 16/100 Superdex 200 column. HA was eluted with 1× PBS. Fractions corresponding to HA was concentrated using Amicon centrifugal filter units with a 30 kDa molecular weight cutoff (Millipore) by centrifugation at 3000×$g$, 4 °C for 10 min.

Glycans were synthesized and printed on microarray slides as previously described[12]. The poly-LacNAc chains were obtained by chemoenzymatic extension using *H. pylori* β1-3 N-acetylglucosaminyltransferase and mammalian β1–4-galactosyltransferase, followed by sialylation by either rat ST3Gal-III or human ST6Gal-I sialyltransferases. Glycans were then custom printed on a MicroGridII (Digilab) using a contact microarray robot equipped with StealthSMP4B microarray pins (Telechem). For analysis, recombinant HA protein (50 μg mL$^{-1}$), anti-His mouse IgG2a antibody (Biolegend, cat no. 362616), and Alexa488-conjugated anti-mouse antibody (Invitrogen, cat no. A28175) were pre-complexed in a 4:2:1 ratio (w/w/w) on ice for 15 min in 100 μL of PBS containing 0.05% Tween-20 (PBS-T). Pre-complexed HAs were incubated on the microarray surface for 60 min in a humidified chamber at room temperature. Slides were washed twice in PBS-T, PBS, then water, and dried until arrays were scanned using an Innoscan 1100AL microarray scanner (Innopsys). Slides contained six replicates of each glycan. Replicates with the highest and lowest fluorescence intensity were excluded, and the fluorescence intensity of the remaining four replicates was used to obtain the mean and standard error of the mean (SEM).

## Synthesis of biotinylated linear and N-linked α-2,6-sialosides

Biotinylated α-2,6-sialosides were synthesized as described previously[12] in which glycan core structures were prepared as formerly described[57]. In brief, biantennary Asn-linked (N-linked) glycans isolated from chicken eggs (GlyTech) were desialylated using *V. cholerae* sialidase, and functionalized to carry a primary amine by coupling NHS-glycine to the Asparagine. The glycans were chemo-enzymatically elongated with one or two Galβ1-4GlcNAc (LacNAc) units using an *H. pylori*, a β1-3 N-acetylglucosamine transferase (β1,3-GlcNAcT), and a recombinant fusion protein comprising a galactosyltransferase and UDP-galactose epimerase from *N. meningitidis* (β4GalT-GalE). Sialylation of terminal galactose to cap glycans in the NeuAcα2-6Gal linkage was accomplished using recombinant human ST6Gal-1. Biotinylation of the glycans was accomplished with NHS-LCLC-biotin and DIPEA. All the biotinylated glycans were confirmed by high-resolution mass spectrometer (HRMS) as follows: 6'SLN$_1$-L, ESI TOF-HRMS m/z calculated for C$_{49}$H$_{84}$N$_7$O$_{23}$S, [M + 2H]$^{2+}$: 585.7709, found 585.7713; 6'SLN$_2$-L, ESI TOF-HRMS m/z calculated for C$_{63}$H$_{107}$N$_8$O$_{33}$S, [M + 2H]$^{2+}$: 768.3370, found 768.3366; 6'SLN$_3$-L, ESI TOF-HRMS m/z calculated for C$_{77}$H$_{130}$N$_9$O$_{43}$S, [M + 2H]$^{2+}$: 950.9031, found 950.9023. HRMS data of 6'SLN$_1$-N, 6'SLN$_2$-N and 6'SLN$_3$-N was reported previously as follows[58]: 6'SLN$_1$-N, ESI TOF-HRMS m/z calculated for C$_{110}$H$_{183}$N$_{12}$O$_{68}$S, [M + 3H]$^{3+}$: 931.0328, found 931.0348; 6'SLN$_2$-N, ESI TOF-HRMS m/z calculated for C$_{138}$H$_{229}$N$_{14}$O$_{88}$S, [M + 3H]$^{3+}$: 1174.4543, found 1174.4567; 6'SLN$_3$-N, ESI TOF-HRMS m/z calculated for C$_{166}$H$_{276}$N$_{16}$O$_{108}$S, [M + 3H]$^{3+}$: 1418.2117, found 1418.2285.

## Measuring glycan binding by ELISA

Streptavidin-coated high binding capacity 384-well plates (Pierce) were washed with PBS 5 times followed by the addition of 50 μL of a 1.8 μM solution of a biotinylated glycan in PBS and incubated at 4 °C overnight. The plates were rinsed with PBS-T 5 times to remove excess glycans. Each well was then blocked with 100 μL of 1% bovine serum albumin (BSA) in PBS containing 0.6 μM desthiobiotin at room temperature for 2 h. Plates were then washed with PBS-T 5 times and used without further processing.

To assess HA binding to the glycan-coated plates, purified His-tagged-HAs (100 μg mL$^{-1}$) were premixed with anti-His mouse IgG2a antibody (Biolegend, cat no. 362616) and HRP-conjugated goat anti-mouse IgG (H + L) secondary antibody (Invitrogen, cat no. G21040) in a 4:2:1 ratio (w/w/w) with 1% BSA in PBS, and incubated on ice for 30 min. The HA complex was then subjected to 3-fold serial dilutions. Fifty microlitre of each premixed HA complex was then transferred to wells of glycan-coated plates and incubated at room temperature for 2 h. The wells were rinsed with PBS-T 5 times and replenished with 50 μL of the 3,3′,5,5′ tetramethylbenzidine (TMB, Sigma-Aldrich, cat no. T0440) peroxidase substrate. The plates were incubated at room temperature for 5 min and then quenched with 50 μL of 2 M sulfuric acid. The absorbance at 450 nm was detected using a BioTek Synergy H1 microplate reader (Agilent). The assay was performed in triplicate.

## Immunization in mouse model

A group of female BALB/c mice aged at 6 weeks old were immunized intraperitoneally with 10,000 PFU of a recombinant virus containing HA and NA from Italy20, and six internal segments from PR8, along with adjuvant (AddaVax, cat no. vac-adx-10). On day 21 post-immunization, a booster was given intraperitoneally using the same amount of adjuvanted virus. On day 21 post-boosting, antisera from mice were obtained from peripheral blood by cardiac puncture and kept at −80 °C until use.

## Human plasma samples

Human plasma samples from 479 healthy adults in Hong Kong in 2021 and 2022 were obtained from our previous study[35]. We selected 28 plasma samples for the present study based on two criteria: (1) the donors were not infected or vaccinated with recent H3N2 virus with HA D186/N190, and (2) plasma samples had 50% microneutralization (MN$_{50}$) titer of at least 1:40 against A/Singapore/INFIMH-16-0019/2016.

## Microneutralization assay

The neutralizing capacity of antisera from immunized mice and plasma samples from human were measured by microneutralization (MN) assay. Briefly, hMDCK cells in a 96-well plate with >90% confluence were washed with PBS once and resupplied with DMEM containing 25 mM HEPES and 100 U mL$^{-1}$ PS. Antisera and plasma samples were heat-inactivated at 56 °C for 30 min and two-fold serially diluted (from 1:10 to 1:1280). Each diluted sample was mixed with 100 TCID$_{50}$ of mutant viruses and incubated at 37 °C for 1 h. The mixture was subsequently incubated with hMDCK cells at 37 °C for 1 h. Cell supernatants were discarded and replaced by DMEM supplemented with 25 mM HEPES, 100 U mL$^{-1}$ PS, and 1 μg mL$^{-1}$ TPCK-trypsin. Plates were incubated at 37 °C for 72 h. The highest dilution that prevented cytopathic effect in at least 50% of the wells was recorded as MN$_{50}$ titer.

## Statistics and reproducibility

Viral titers, neutralizing titers, optical density 450 nm values and melting temperatures were measured with at least three independent biological replicates. Wilcoxon signed-rank test was used to analyze the statistical differences of neutralizing titers between groups. No data were excluded from the analyses.

## Reporting summary

Further information on research design is available in the Nature Portfolio Reporting Summary linked to this article.

## Data availability

Raw sequencing data have been deposited to the NIH Short Read Archive under accession number: PRJNA883249. The X-ray coordinates and structure factors have been deposited in the RCSB Protein Data Bank under accession codes 8FAQ and 8FAW. Structures from the following identifiers from the Protein Data Bank (PDB) were used in this study: 2YP2, 2YP7, 4FNK, 4O5N, 6AOQ, 6BKP, 6TZB, and 8TJA. Source data are provided with this paper.

## Code availability

Custom codes to process next-generation sequencing data can be accessed at https://doi.org/10.5281/zenodo.11099315.

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

## Acknowledgements

This work was supported by a fellowship from the Pasteur Foundation Asia (W.L.), InnoHK, an initiative of the Innovation and Technology Commission, the Government of the Hong Kong SAR (R.B.), the Searle Scholars Program (N.C.W.), the Howard Hughes Medical Institute Emerging Pathogens Initiative (N.C.W.), the Health and Medical Research Fund (No. 19180932) (C.K.P.M.), Emergency Key Program of Guangzhou Laboratory (Grant No. EKPG22-30-6) (C.K.P.M.), and the visiting scientist scheme from Lee Kong Chian School of Medicine, Nanyang Technological University, Singapore (C.K.P.M.).

## Author contributions

R.L., W.L., W.O.O., and A.H.G. equally contributed to this study. R.L., W.L., W.O.O., R.B., J.C.P., S.K.N., C.K.P.M., and N.C.W. conceived and designed the study. R.L., W.L., W.O.O., A.H.G., C.K., S.W., R.M., Y.S., C.C., T.J.C.T., C.S.G., L.A.R., I.R.S., and D.C. performed the experiments. R.L., W.L., W.O.O., C.K., and S.W. analyzed the data. R.B., J.C.P., S.K.N., C.K.P.M., and N.C.W. provided resources and support. R.L., W.L., W.O.O., C.K.P.M., and N.C.W. wrote the paper, and all authors reviewed and edited the paper.

## Competing interests

N.C.W. consults for HeliXon. The authors declare no other competing interests.
