## [Peer Review File · Nature Communications]

REVIEWER COMMENTS

Reviewer #1 (Remarks to the Author):

In this manuscript, Lei and colleagues probe the functional consequences of recent amino acid substitutions in the hemagglutinin (HA) of recent H3N2 viruses. In particular, the authors extend their previous body of work on epistatic interactions in the HA receptor-binding domain (RBD) to two residues that recently underwent convergent evolution (G186D and D190N). Using a combination of structural biology, glycan binding analyses, and deep mutational scanning, the authors show that while the individual D186 and N190 mutations are deleterious for binding to sialylated glycans, their combination restores affinity and shifts the binding mode. Further, using deep mutational scanning the authors demonstrate that these residues are broadly incompatible with a previously identified egg-adaptive RBD substitution, L194P, suggesting broader epistatic interactions within the RBD.

In general, the data are high-quality and convincingly demonstrate the epistatic interactions between G186D and D190N. The manuscript is also well written and for the most part easy to follow. There are some areas, however, where changes to the text and presentation of data could help improve the reader's understanding.

Major points:

1. While the flow of experiments for demonstrating epistasis between G186D and D190N is logical, the authors should expand more on the rationale for conducting deep mutational scanning experiments to assess compatibility with L194P specifically. While this is a previously-identified egg-adaptive change, the significance of this isn't clear to a reader who isn't familiar with some of the relevant literature. For example, is the rationale to see the likelihood of the egg-adaptive L194P change emerging on a background containing G186P and D190N? Giving the reader some context of why L194P was specifically chosen for further study would be helpful.
2. The authors note that the double mutation of G186D and D190N has evolved in two independent instances, however there are other differences between 3C.2a1b.1a and 3C.2a1b.2a2 as well. One of these residues includes T160I, which is interesting as T160 was also incompatible with L194P and I160 results in the loss of an N-linked glycan in close proximity to the RBD that also had consequences for receptor-binding avidity. The authors might comment on this given that it is possible T160I is involved in epistatic interactions as well (and 3C.2a1b.2a2 seems to be dominating circulation).

3. While the mouse immunization experiments show that these mutations likely alter antigenicity, the more relevant question for escape from population immunity is whether sera from humans or animals exposed to previous H3N2 strains containing G186 and D190 has reduced neutralization capacity against viruses containing D186 and N190. If available, these data could help shed some light on the magnitude of the antigenic change due to these two mutations.

Minor points:

1. Line 86: probably better said as “These two human H3N2 clades are distinguished by” or “These two human H3N2 clades share the HA mutation D190N”

Reviewer #2 (Remarks to the Author):

This manuscript describes the epistatic phenotypes of positions 186 and 190 in contemporary H3N2 viruses. It is based on numerous previous observations of Nicholas Wu over the past decade and is a natural analysis of the continued evolution of H3N2 viruses. The data is solid and presented nicely. It is however stated two times (L111 & L230) in the manuscript that the changes in 186 and 190 are the second time that the receptor binding site is “evolved”. H3N2 antigenic and receptor binding evolution is quite gradual, with differences in antigenic clusters (vaccine strains and mismatches) and receptor binding specificity (1, 2 or 3 LacNAc repeats). Because this manuscript is such a continuation both the introduction and discussion are extremely short, and it comes across of just putting the data together, the paper would be much improved if given more substance.

I would expect that in every couple of years, 1 to 3 mutations will alter antigenicity and receptor binding properties. For example, the L194P mutation was an egg adaptation that caused a major antigenic shift and was incompatible with receptor binding; fortunately, this mutation appeared to be incompatible in later (vaccine)strains. Thus, to determine which amino acids are responsible for this incompatibility is admirable but the importance is hardly described. Finally, the statement that D190N in 3c3 viruses results in avian-type receptor binding is not shown in the reference used here, or somewhere else to my knowledge. An egg adaptation in the RBS is not an avian-type receptor mutant perse.

Some important points for consideration:

- The statement that the antigenic drift affects the receptor binding mode and specificities uses 8 references, some of them are rather dated. Recent literature describes the relationship between

antigenicity and receptor binding in detail. For example, the Medeiros reference on the inability of H3N2 viruses to bind erythrocytes is explained in PMID: 34521834.

- The introduction is extremely short, the 2nd paragraph describes that the L194P egg adaptation cannot occur in contemporary strains. Isn't this a good thing? Or bad because the virus doesn't replicate in eggs anymore? what is the importance of this study for a broader audience?
- Both the introduction and the discussion need additional substance regarding receptor binding and antigenic evolution.
- L104 the authors immediately state that the receptor binding mode is different due to positions 186 and 190, but if I analyze fig 1c why do positions 225, 193 and 135 not affect the binding mode?
- Figure 2 Panel 2 could be transformed to a simple table to demonstrate the non-occurrence of DD and GN. For panel b please add the amino acids letters within the residue panels.
- The analyses of receptor binding properties are nice but should be discussed regarding the symmetry of the structures used.
- L161 stability effects is dropped here without any introduction, nor was it in the title of this paragraph. Also, that this is not simply cumulative is not surprising.

Minor

L60A viruses do not grow, but replicate

L61 receptor specificity adapts to

L77 "weakened" binding?

L81 The MS is about H3, so H3 numbering seems logical to me.

L121 refers to figure 2, should be 1f-g

RESPONSE TO REVIEWERS' COMMENTS

Reviewer #1 (Remarks to the Author):

In this manuscript, Lei and colleagues probe the functional consequences of recent amino acid substitutions in the hemagglutinin (HA) of recent H3N2 viruses. In particular, the authors extend their previous body of work on epistatic interactions in the HA receptor-binding domain (RBD) to two residues that recently underwent convergent evolution (G186D and D190N). Using a combination of structural biology, glycan binding analyses, and deep mutational scanning, the authors show that while the individual D186 and N190 mutations are deleterious for binding to sialylated glycans, their combination restores affinity and shifts the binding mode. Further, using deep mutational scanning the authors demonstrate that these residues are broadly incompatible with a previously identified egg-adaptive RBD substitution, L194P, suggesting broader epistatic interactions within the RBD.

In general, the data are high-quality and convincingly demonstrate the epistatic interactions between G186D and D190N. The manuscript is also well written and for the most part easy to follow. There are some areas, however, where changes to the text and presentation of data could help improve the reader's understanding.

Response: Thank you for the encouraging comment.

Major points:

1. While the flow of experiments for demonstrating epistasis between G186D and D190N is logical, the authors should expand more on the rationale for conducting deep mutational scanning experiments to assess compatibility with L194P specifically. While this is a previously-identified egg-adaptive change, the significance of this isn't clear to a reader who isn't familiar with some of the relevant literature. For example, is the rationale to see the likelihood of the egg-adaptive L194P change emerging on a background containing G186P and D190N? Giving the reader some context of why L194P was specifically chosen for further study would be helpful.

Response: Thank you for your constructive suggestions. In the revised manuscript, we have provided more context on why L194P was chosen for study:

Lines 68-78: "Certain egg-adaptive mutations can alter HA antigenicity and hence hamper vaccine efficacy²⁰⁻²³, as exemplified by L194P mutation²³⁻²⁵. While L194P was a common egg-adaptive mutation in egg-grown H3N2 vaccine strains prior to 2020, it is rarely observed in recent years²⁶. Consistently, our previous study has shown that L194P is incompatible with recent human H3N2 strains²⁶. In other words, L194P imposes a huge replication fitness cost to human H3N2 strains from recent years but not before 2020. This observation demonstrates that epistasis exists between L194P and recently emerged natural mutations in human H3N2 HA. It also suggests that recent evolution of human H3N2 virus involves structural changes in the HA RBS. Given that most seasonal influenza vaccines are still produced in eggs and different egg-adaptive mutations have different antigenic effects²⁷, it is critical to understand of how natural mutations in human H3N2 HA influence the preference of egg-adaptive mutations."

Lines 214-218: "Before 2020, L194P is a common egg-adaptive mutation that alters the HA antigenicity of egg-grown H3N2 vaccine strains^{24,26}. Nevertheless, L194P is rarely observed in recent egg-grown H3N2 vaccine strains, due to epistatic interactions with

natural mutations that emerged during HA RBS evolution²⁶. Such epistatic interactions lead to an incompatibility between L194P and recent human H3N2 strains.”

2. The authors note that the double mutation of G186D and D190N has evolved in two independent instances, however there are other differences between 3C.2a1b.1a and 3C.2a1b.2a2 as well. One of these residues includes T160I, which is interesting as T160 was also incompatible with L194P and I160 results in the loss of an N-linked glycan in close proximity to the RBD that also had consequences for receptor-binding avidity. The authors might comment on this given that it is possible T160I is involved in epistatic interactions as well (and 3C.2a1b.2a2 seems to be dominating circulation).

Response: Thank you for this constructive comment. To understand whether T160I is involved in any epistatic interactions, we have included additional mutagenesis experiments in the revised manuscript. Results are now presented in Supplementary Figure 3 and elaborated in results section:

Lines 251-259: “As mentioned above, 3C.2a1b.1a and 3C.2a1b.2a2 are the two recent human H3N2 clades with G186D and D190N. Unlike clade 3C.2a1b.1a, which has an N-glycosylation site at residue 158, clade 3C.2a1b.2a2 has lost this N-glycosylation site via mutation T160I (Supplementary Fig 3a)⁶. Given that T160K helped restore L194P compatibility in Italy20, we aimed to further test if T160I could do the same by replacing T160K in Mut7 by T160I. Our rescue experiment showed that Mut7 with T160I instead of T160K failed to compensate the fitness defect of Italy20-L194P (Supplementary Fig 3b), despite both mutations led to a loss of an N-glycosylation site at residue 158. This observation suggests that I160 may also contribute to the L194P incompatibility in clade 3C.2a1b.2a2 through a mechanism unrelated to N-glycosylation.”

3. While the mouse immunization experiments show that these mutations likely alter antigenicity, the more relevant question for escape from population immunity is whether sera from humans or animals exposed to previous H3N2 strains containing G186 and D190 has reduced neutralization capacity against viruses containing D186 and N190. If available, these data could help shed some light on the magnitude of the antigenic change due to these two mutations.

Response: We agree with the reviewer that such data are informative. In the revised manuscript, we have tested the neutralizing titers of plasma samples from 28 human participants with pre-existing immunity to pre-2020 human H3N2 strains (G186/D190) against Italy20 WT (D186/N190) and different Italy20 mutants. This result is presented in Figure 4b and described in the results section of the revised manuscript:

Lines 204-211: “We further measured the neutralizing activities of plasma samples from human who had not been infected or vaccinated by recent H3N2 strains with D186/N190³⁵. As expected, these plasma samples had significantly higher MN₅₀ titers against an older H3N2 strain in clade 3C.2a, namely A/Singapore/INFIMH-16-0019/2016 (Sing16), than Italy20 WT ($P < 0.05$, paired Wilcoxon signed-rank test) (Fig 4c). Specifically, the average MN₅₀ titer decreased from above 1:160 against Sing16 to below 1:20 against Italy20 WT. Such decrease in MN₅₀ titers could be partly compensated to around 1:80 by D186G ($P < 0.05$, paired Wilcoxon signed-rank test, Fig 4c). This observation suggests that G186D is involved in the antigenic drift of human H3N2 virus.”

Details of these human serum samples are described in the methods section of the manuscript:

Lines 519-525: “Human plasma samples from 479 healthy adults in Hong Kong in 2021 and 2022 were obtained from our previous study³⁵, which was approved by the Joint Chinese University of Hong Kong-New Territories East Cluster (Ref no: 2020.229) Clinical Research Ethics Committee. We selected 28 plasma samples for the present study based on two criteria: 1) the donors were not infected or vaccinated with recent H3N2 virus with HA D186/N190, and 2) plasma samples had 50% microneutralization (MN₅₀) titer of at least 1:40 against A/Singapore/INFIMH-16-0019/2016.”

Minor points:

1. Line 86: probably better said as “These two human H3N2 clades are distinguished by” or “These two human H3N2 clades share the HA mutation D190N”

Response: Thank you for the suggestion. The sentence is rephrased as suggested:

Lines 94: “These two human H3N2 clades share the HA mutation D190N ...”

Reviewer #2 (Remarks to the Author):

This manuscript describes the epistatic phenotypes of positions 186 and 190 in contemporary H3N2 viruses. It is based on numerous previous observations of Nicholas Wu over the past decade and is a natural analysis of the continued evolution of H3N2 viruses. The data is solid and presented nicely. It is however stated two times (L111 & L230) in the manuscript that the changes in 186 and 190 are the second time that the receptor binding site is “evolved”. H3N2 antigenic and receptor binding evolution is quite gradual, with differences in antigenic clusters (vaccine strains and mismatches) and receptor binding specificity (1, 2 or 3 LacNAc repeats). Because this manuscript is such a continuation both the introduction and discussion are extremely short, and it comes across of just putting the data together, the paper would be much improved if given more substance.

Response: Thank you for the constructive comments. We agree that it is misleading to state that mutations at positions 186 and 190 represent the second instance of the receptor binding mode evolving. Those sentences have been modified in the revised manuscript:

Lines 123-125 (previously lines 111-113): “While the receptor binding mode of human H3N2 HA has evolved throughout the past decades^{2,8,11,24}, our structural analysis here demonstrates that this evolution continues in recent human H3N2 HA.”

Lines 265-266 (previously lines 230-232): “The HA receptor binding mode of human H3N2 virus has been evolving since its emergence in 1968^{2,8,11,24,29}.”

Besides, the introduction and discussion are expanded to provide more substance (see responses below).

I would expect that in every couple of years, 1 to 3 mutations will alter antigenicity and receptor binding properties. For example, the L194P mutation was an egg adaptation that caused a major antigenic shift and was incompatible with receptor binding; fortunately, this mutation appeared to be incompatible in later (vaccine)strains. Thus, to determine which amino acids are responsible for this incompatibility is admirable but the importance is hardly described. Finally, the statement that D190N in 3c3 viruses results in avian-type receptor binding is not shown in the reference

used here, or somewhere else to my knowledge. An egg adaptation in the RBS is not an avian-type receptor mutant perse.

Response: In the revised manuscript, we have provided more context on why L194P was chosen for study (see above response to the first major comment from reviewer #1).

As for D190N, we agree with the reviewer that an egg adaptive mutation in the RBS does not necessarily mean that it facilitates binding to avian-type receptor. As a result, we have rephrased our discussion paragraph on D190N accordingly:

Lines 285-295: "Previous studies have shown that D190N occurs in egg-grown but not circulating human H3N2 strains from clade 3C.3a^{26,40}. It is unclear how D190N acts as an egg-adaptive mutation and at the same time emerges in human H3N2 virus with strong binding preference for human-type receptor. One possibility is that D190N, albeit being an egg-adaptive mutation, does not facilitate binding to avian-type receptor. Another mutually non-exclusive possibility is that the effect of D190N on receptor specificity is strain-dependent. We postulate that the latter is likely, given that most egg-adaptive mutations are known to increase binding avidity to avian-type receptor^{18,21,23,24,41} and epistatic interactions are prevalence in the HA RBS^{11,29,32}. Nevertheless, this perplexing observation of D190N being both an egg-adaptive mutation and a natural mutation in circulating strains highlights that the sequence determinants of HA receptor specificity remain to be fully characterized."

Some important points for consideration:

- The statement that the antigenic drift affects the receptor binding mode and specificities uses 8 references, some of them are rather dated. Recent literature describes the relationship between antigenicity and receptor binding in detail. For example, the Medeiros reference on the inability of H3N2 viruses to bind erythrocytes is explained in PMID: 34521834.

Response: Thank you for bringing up this suggestion. More recent studies (PMID: 34521834 and PMID: 38307019) are now cited throughout the manuscript.

- The introduction is extremely short, the 2nd paragraph describes that the L194P egg adaptation cannot occur in contemporary strains. Isn't this a good thing? Or bad because the virus doesn't replicate in eggs anymore? what is the importance of this study for a broader audience?

Response: In the introduction of the revised manuscript, we have provided more context on why L194P was chosen for study (see above response to the first major comment from reviewer #1). This includes a sentence that states the public health implications of understanding epistasis between natural mutations and egg-adaptive mutations.

Lines 75-78: "Given that most seasonal influenza vaccines are still produced in eggs and different egg-adaptive mutations have different antigenic effects²⁷, it is critical to understand of how natural mutations in human H3N2 HA influence the preference of egg-adaptive mutations."

While L194P cannot occur in recent human H3N2 strains, it is unclear to us whether it is a good or bad thing. As shown in our previous study (PMID: 36155668), many recent human H3N2 strains adapt to eggs via another egg-adaptive mutation D225G. Although residue 225 is in the major antigenic site D, the antigenic effect of D225G on recent human H3N2 strains remains largely

elusive. Therefore, we prefer not to comment on whether the incompatibility between L194P and recent human H3N2 strains is good or not.

- Both the introduction and the discussion need additional substance regarding receptor binding and antigenic evolution.

Response: The introduction is expanded to provide more background information regarding receptor binding and antigenic evolution:

Lines 57-64: “When H3N2 virus first emerged in human, its HA receptor specificity switched from α 2,3-linked sialylated glycan (avian-type receptor) to α 2,6-linked sialylated glycan (human-type receptor)^{14,15}. In the past two decades, human H3N2 virus has further evolved its HA receptor binding specificity towards a subset of α 2,6-linked sialylated glycans that are branched and have extended poly-N-acetyl-lactosamine chains¹². This evolution of receptor specificity is attributed to HA RBS mutations that form additional interactions beyond the terminal sialic acid (Sia-1)^{2,16}. Many of these mutations are also associated with antigenic drift^{2,17}.”

Similarly, we have also discussed our finding of D190N-driven receptor binding evolution in the discussion section:

Lines 274-283: “HA residue 190 locates at the center of the RBS and has three major variants in human H3N2 virus. It was a Glu when human H3N2 virus first emerged in human in 1968, mutated to Asp during the 1992–1993 influenza season, and mutated again to Asn in two recent clades 3C.2a1b.1a and 3C.2a1b.2a2 (Fig 2a-b). As clade 3C.2a1b.2a2 was the predominant clade during the 2023–2024 influenza season, Asn has reached fixation²⁸. E190 in early human H3N2 virus H-bonds with Sia-1^{29,39}. D190 in human H3N2 virus from 2000s to 2010s forms water-mediated H-bonds with both Sia-1 and GlcNAc-3². As shown in this study, N190 in recent human H3N2 virus directly H-bonds with GlcNAc-3 without interacting with Sia-1. Thus, the evolution of residue 190 is consistent with the notion that human H3N2 HA RBS tends to build up interaction along the length of the receptor glycan chain rather than the terminal Sia-1².”

- L104 the authors immediately state that the receptor binding mode is different due to positions 186 and 190, but if I analyze fig 1c why do positions 225, 193 and 135 not affect the binding mode?

Response: We have updated the structural comparison by replacing A/Brisbane/10/2007 (Bris07) HA + 6'SLNLN with Ecu16 (A/Ecuador/1374/2016) HA + 6'SLNLN, which is our recently published structure (PMID: 38307019). This update improves the analysis of the evolution of receptor binding mode, since Ecu16 is only 4 years older than Vic20, while Bris07 is 13 years older. Thus, the receptor binding site of Ecu16 has a higher sequence identity to that of Vic20. Ecu16 HA and Vic20 HA have the same amino acid sequences at positions 193 and 225. The only differences in the HA RBS of Ecu16 and Vic20 are positions 135, 186, and 190. While Ecu16 and Vic20 have different receptor binding modes, we do not think position 135 affect the binding mode due to the following reason described in the revised manuscript:

Lines 113-116: “Ecu16 HA and Vic20 HA differed by three mutations in the RBS, namely T135K, G186D, and D190N (Fig 1d-e). Since position 135 only interacted with the Sia-1 moiety of the human-type receptor analog, which had an almost identical positioning in Ecu16 HA and Vic20 HA, T135K did not seem to have a major role in the evolution of receptor binding mode.”

• Figure 2 Panel 2 could be transformed to a simple table to demonstrate the non-occurrence of DD and GN. For panel b please add the amino acids letters within the residue panels.

Response: Thank you for your suggestion. Amino acids are now indicated along with the corresponding frequency lines in panel b. For panel a, we believe that a phylogenetic tree provides key information that cannot be presented by a table. Specifically, a phylogenetic tree, but not a table, shows the co-occurrence of G186D and D190N in two antigenic distinct clades.

• The analyses of receptor binding properties are nice but should be discussed regarding the symmetry of the structures used.

Response: We have added the following sentences in the revised manuscript to discuss the symmetry of different structures used:

Lines 136-139: “This difference is unlikely to be an artifact of crystal packing, because the RBS is not involved in crystal packing interface of these HA structures. Additionally, the HA apo structures of Fin04, HK05, Bris07, Vic11, Mich14, and Vic20 all belong to the space group H 3 2 and have the same crystal packing.”

• L161 stability effects is dropped here without any introduction, nor was it in the title of this paragraph. Also, that this is not simply cumulative is not surprising.

Response: The subsection title of this paragraph is revised to: “Receptor binding and thermostability analyses reveal epistasis between G186D and D190N”.

Minor

L60A viruses do not grow, but replicate

Response: We have replaced “grows” by “replicates” in line 66 (previously line 60).

L61 receptor specificity adapts to

Response: We have replaced “switches” by “adapts” in line 67 (previously line 61).

L77 “weakened” binding?

Response: This is rephrased to “... weakened this binding avidity” in line 86 (previously line 77) to improve clarity.

L81 The MS is about H3, so H3 numbering seems logical to me.

Response: We have removed the sentence about H3 numbering, because it sounds redundant.

L121 refers to figure 2, should be 1f-g

Response: Thanks for catching this typo, which is corrected in line 133 (previously line 121).

REVIEWERS' COMMENTS

Reviewer #1 (Remarks to the Author):

In a revised manuscript, Lei & colleagues include new data measuring the effects of epistatic RBD mutations on antigenicity in the context of human serum samples, as well as structural comparisons using a more recent H3N2 strain. The authors also show incompatibility of K160I with the egg-adaptation L194P, further demonstrating epistatic interactions between RBD residues. The authors were responsive to reviewer comments and the results and discussion of the manuscript better situate the findings in the context of the field and previous work.

A few minor comments:

1. The manuscript might benefit from a sentence or two at the end of the discussion that summarizes the findings – currently it ends a bit abruptly.
2. Line 87: “alters the HA antigenicity”
3. Line 274: might be better to phrase as “HA residue 190 is located at the center of the RBS and three major amino acids have been observed in human H3N2 viruses”
4. Line 282: “tends” might not be the best word – might be better to say “has evolved additional contacts”

Reviewer #1 (Remarks on code availability):

I lack the technical background to analyze the code

Reviewer #2 (Remarks to the Author):

The authors edited their manuscript nicely; many thanks.

RESPONSE TO REVIEWERS' COMMENTS

Reviewer #1 (Remarks to the Author):

In a revised manuscript, Lei & colleagues include new data measuring the effects of epistatic RBD mutations on antigenicity in the context of human serum samples, as well as structural comparisons using a more recent H3N2 strain. The authors also show incompatibility of K160I with the egg-adaptation L194P, further demonstrating epistatic interactions between RBD residues. The authors were responsive to reviewer comments and the results and discussion of the manuscript better situate the findings in the context of the field and previous work.

Response: Thank you for the positive comment.

Minor points:

1. The manuscript might benefit from a sentence or two at the end of the discussion that summarizes the findings – currently it ends a bit abruptly.

Response: Thank you for the suggestion. Two additional sentences are now added after the discussion.

Lines 307-309: “Due to the public health relevance of HA RBS evolution, continued studies of its functional constraints and the underlying biophysical basis are warranted.”

2. Line 87: “alters the HA antigenicity”

Response: Fixed

3. Line 274: might be better to phrase as “HA residue 190 is located at the center of the RBS and three major amino acids have been observed in human H3N2 viruses”

Response: The sentence is revised as suggested in lines 275-276.

4. Line 282: “tends” might not be the best word – might be better to say “has evolved additional contacts”

Response: This sentence is revised as suggested in lines 283-284.

Reviewer #1 (Remarks on code availability):

I lack the technical background to analyze the code

Response: The codes provided in Code Availability section have been repeatedly checked to ensure the reproducibility of results.

Reviewer #2 (Remarks to the Author):

The authors edited their manuscript nicely; many thanks.

Response: Thank you for your encouraging comments.